# SARS-CoV-2 Specific Antibody Response and T Cell-Immunity in Immunocompromised Patients up to Six Months Post COVID: A Pilot Study

**DOI:** 10.3390/jcm11123535

**Published:** 2022-06-20

**Authors:** Johanna Sjöwall, Maria Hjorth, Annette Gustafsson, Robin Göransson, Marie Larsson, Hjalmar Waller, Johan Nordgren, Åsa Nilsdotter-Augustinsson, Sofia Nyström

**Affiliations:** 1Department of Biomedical and Clinical Sciences, Division of Inflammation and Infection, Linköping University, SE-581 85 Linköping, Sweden; annette.gustafsson@regionostergotland.se (A.G.); asa.nilsdotter-augustinsson@liu.se (Å.N.-A.); 2Department of Clinical Immunology and Transfusion Medicine, and Department of Biomedical and Clinical Sciences, Linköping University, SE-581 85 Linköping, Sweden; maria.hjorth@liu.se (M.H.); robin.goransson@regionostergotland.se (R.G.); sofia.c.nystrom@liu.se (S.N.); 3Department of Biomedical and Clinical Sciences, Division of Molecular Medicine and Virology, Linköping University, SE-581 85 Linköping, Sweden; marie.larsson@liu.se (M.L.); hjawa871@student.liu.se (H.W.); johan.nordgren@liu.se (J.N.); 4Department of Infectious Diseases, University Hospital, SE-581 85 Linköping, Sweden

**Keywords:** COVID-19, immunocompromised, SARS-CoV-2 specific T cell responses, SARS-CoV-2 IgG, viral load, immunity

## Abstract

COVID-19 generates SARS-CoV-2-specific antibodies in immunocompetent individuals. However, in immunocompromised patients, the humoral immunity following infection may be impaired or absent. Recently, the assessment of cellular immunity to SARS-CoV-2, both following natural infection and vaccination, has contributed new knowledge regarding patients with low or no antibody responses. As part of a prospective cohort study which included hospitalized patients with COVID-19, we identified immunocompromised patients and compared them with age- and sex-matched immunocompetent patients regarding co-morbidities, biomarkers of COVID-19 and baseline viral load by real-time PCR in nasopharyngeal swabs. Spike and nucleocapsid antibody responses were analyzed at inclusion and after two weeks, six weeks and six months. Plasma immunoglobulin G (IgG) levels were quantified, lymphocyte phenotyping was performed, and SARS-CoV-2 specific CD4 and CD8 T cell responses after in vitro antigen stimulation were assessed at six months post infection. All patients showed IgG levels above or within reference limits. At six months, all patients had detectable SARS-CoV-2 anti-spike antibody levels. SARS-CoV-2 specific T cell responses were detected in 12 of 12 immunocompetent patients and in four of six immunocompromised patients. The magnitude of long-lived SARS-CoV-2 specific T cell responses were significantly correlated with the number of CD4 T cells and NK cells. Determining the durability of the humoral and cellular immune response against SARS-CoV-2 in immunocompromised individuals could be of importance by providing insights into the risk of re-infection and the need for vaccine boosters.

## 1. Introduction

The ongoing pandemic caused by the severe acute respiratory syndrome coronavirus 2 (SARS-CoV-2) has particularly concerned and affected patients with certain co-morbidities, not least those with various immunosuppressive disorders and therapies who have an increased risk of developing severe coronavirus disease 2019 (COVID-19) [1,2,3,4,5]. Several factors are likely to contribute to the increase in risk. Inability of viral clearance in certain immunodeficient states, resulting in hyperinflammation and progression into acute respiratory distress syndrome, has been described [6]. Moreover, an impaired SARS-CoV-2 specific antibody response due to natural infection or vaccination to SARS-CoV-2 especially in patients with solid organ transplants, autoimmune diseases, and haematological cancers may contribute to the increase in morbidity and mortality due to COVID-19 [7,8,9,10,11,12,13,14]. Antibody levels in SARS-CoV-2 seropositive, immunocompromised patients have also been shown to decline more rapidly than in immunocompetent individuals [15], although the longevity and function of the antibodies is not completely known. Although neutralizing antibodies are considered the most important surrogate marker for immunity to SARS-CoV-2, accumulating evidence suggests that cellular responses may also play an important role in preventing the initial infection and development of severe disease [16]. Interestingly, despite various antibody levels, an identical SARS-CoV-2 specific T cell response rate following SARS-CoV-2 vaccination was observed in immunocompromised and immunocompetent individuals [17]. However, due to their functional heterogeneity in different immunodeficient conditions, the short- and long-term protective role of T cells in COVID-19 is still incompletely understood. 

Thus, herein we aimed to explore the longitudinal antibody response and long-lived specific T cell response to SARS-CoV-2 in patients with various immunosuppressive disorders that were hospitalized due to COVID-19.

## 2. Materials and Methods

### 2.1. Study Design

This pilot study is a part of a prospective, observational cohort study described elsewhere [18], implemented during August 2020 to May 2021, and involving hospitalized adult patients with COVID-19. 

### 2.2. Patients and Blood Sampling 

From the original cohort (*n* = 62) of hospitalized patients with COVID-19, we identified six patients (two immunocompromised patients from the original cohort were excluded; one patient with acute myeloid leukemia died during the study period and one with multiple myeloma was too ill to participate in the six-month sampling) with immunosuppressive disorders (henceforth referred to as immunocompromised) and compared them with 12 immunocompetent patients matched for age and gender. The immunocompromised patients had medical conditions affecting the immune system and/or received immunosuppressive therapy, whereas the immunocompetent patients had none of these conditions. Clinical characteristics and COVID-associated therapy were obtained from questionnaires and medical records. Routine biochemical variables in peripheral blood were analyzed at inclusion.

### 2.3. Real-Time PCR for Quantification of SARS-CoV-2 Viral Load

At admittance to hospital, and as part of routine diagnostics, all patients and controls had COVID-19 verified by Abbott Real Time SARS-CoV-2 or Alinity m SARS-CoV-2 assays (Abbott, Solna, Sweden) using nasopharyngeal or throat swab specimens, performed at the Clinical Microbiology Laboratory, Linköping University Hospital, Sweden. Nasopharyngeal samples were resampled from patients and controls at study inclusion (at a later moment when admitted to the Department of Infectious Diseases, Vrinnevi Hospital, Norrköping, Sweden) and kept at −80 °C until analysis. RNA extraction was performed using the QIAamp Viral RNA Mini Kit (QIAGEN, 52904) according to the manufacturer’s instructions. A High-Capacity cDNA Reverse Transcription Kit (Thermo Fisher Scientific, 4368814, Waltham, MA, USA) was used for cDNA synthesis, following the manufacturer’s instructions. The real-time PCR (qPCR) for the SARS-CoV-2 envelope gene was performed according to Hagbom et al. [19]. In short, all samples were run in duplicate on a CFX96 (Bio-Rad, Solna, Sweden) using iTaq Universal Probes Supermix (Bio-Rad) with primers and probe according to Corman et al. [20]. qPCR negative samples were given a ct-value of 39 (corresponding to half of the in-house determined detection limit) for statistical purposes.

### 2.4. Analysis of Longitudinal SARS-CoV-2 IgG Antibodies

Quantification of anti-SARS-CoV-2 spike (S) IgG and anti-SARS-CoV-2 nucleocapsid (N) IgG was performed at inclusion, at two and six weeks, and at six months post infection, respectively, at the Department of Clinical Microbiology, Linköping, Sweden, according to standard operating procedures. Chemiluminescent microparticle-based immune assays were used for quantification of anti-S IgG directed against the receptor binding domain (RBD) of the spike protein (Architecht SARS-CoV-2 II Quant, 6S60), and anti-N IgG (Architect, SARS-CoV-2 IgG, 6R86) assays (Abbott Laboratories Diagnostics Division, Abbott Scandinavia AB). The anti-S IgG RBD assay was calibrated to the WHO International Reference anti-SARS IgG antibody NIBSC 20-136. Linear correlation with neutralization activity has been demonstrated for the anti-S IgG RBD assay at levels >17 binding arbitrary units (BAU)/mL [21]. The anti-SARS-CoV-2 S IgG positivity cut-off was 14.2 BAU/mL. The anti-SARS-CoV-2 N IgG positivity cut-off was 1.4 AU/mL.

### 2.5. Lymphocyte Phenotyping by Flow Cytometry

Absolute T, B and NK cell numbers were determined in peripheral blood by flow cytometry using Trucount tubes (cat no 663028, BD Biosciences, San Jose, CA, USA), in all patient samples six months post infection. FITC-conjugated anti-CD3, PE-conjugated anti-CD8, PerCP-conjugated anti-CD45 and APC-conjugated anti-CD4 antibodies were used with BD Multitest (cat no 340499, BD Biosciences), and anti-CD3, anti-CD45, APC-conjugated anti-CD19 and PE-conjugated anti-CD16/56 with BD Multitest (cat 340500). Briefly, antibodies were added to whole blood and samples were lysed with FACS lysing solution (BD Biosciences). Data were acquired using FACS Canto II (BD Biosciences). Kaluza flow cytometry software version 2.1 (Beckman Coulter, Indianapolis, IN, USA) was used for data analysis. The gating strategy is available in Appendix A.

### 2.6. T Cell Mediated Anti-SARS-CoV-2 Specific Immune Responses

SARS-CoV-2 specific T cells were detected by flow cytometric assay for antigen specific cell-mediated recall immune responses in activated whole blood in all patient samples six months post infection. Sodium-heparinized whole blood was diluted 1:10 in culture medium (RPMI 1640; Gibco, Paisley, UK and incubated for seven days with immunogenic class I peptide overlapping pools of the SARS-CoV-2 S-proteins and N-proteins, respectively (cat no 130-126-701;130-126-699, Miltenyi Biotec, Bergisch Gladbach, Germany). Stimulation with phytohemagglutinin (PHA; Roche Diagnostics GmbH, Mannheim, Germany) was used as positive control and culture medium was used as negative control. Expanded blast transformed CD4 and CD8 T cells were quantified by flow cytometry and the use of Trucount tubes. Data were acquired using FACS Canto II (BD Biosciences) and Kaluza flow cytometry software version 2.1 (Beckman Coulter) was used for data analysis. 

### 2.7. Statistical Analyses

Statistical calculations and graphical illustrations were made with Graphpad Prism version 9.1.2 (GraphPad, La Jolla, CA, USA). Mann-Whitney *U* test and Fischer’s exact test were used for comparison between groups and Spearman’s Rank test was used for correlations. *p*-values <0.05 were considered statistically significant.

## 3. Results

### 3.1. Clinical Characteristics of the Study Participants

Patient characteristics, COVID-associated therapy, biochemical variables, and SARS-CoV-2 serology at inclusion are presented in Table 1. The immunocompromised patients (*n* = 6, 50% men, median age 62 years) were diagnosed with haematological malignancies (*n* = 2; pomalidomide plus dexamethasone treated myeloma and untreated chronic lymphocytic leukemia, respectively) and spondyloarthritis (*n* = 2; both treated with TNF blocking agents and sulfasalazine). In addition, two patients were kidney transplant recipients (treated with mycophenolic acid/cyclosporine/tacrolimus/corticosteroids). No statistically significant differences in clinical characteristics, therapy, biochemical variables, or SARS-CoV-2 antibodies (at inclusion) were observed between the immunocompromised and the immunocompetent patients. None in the study cohort were re-infected with SARS-CoV-2 within six months of the last follow-up (one year from study inclusion), but the patient with myeloma died one month after the six-month study visit.

### 3.2. SARS-CoV-2 Viral Load

At study inclusion, five of six (83%) immunocompromised patients had detectable levels of SARS-CoV-2 RNA in nasopharyngeal samples compared to four of twelve (33%) immunocompetent patients (Appendix A). This is explained by the fact that some of the patients were initially receiving care at another pandemic department and were not included in the study until they were transferred to the Department of Infectious Diseases. The immunocompromised patients had a trend towards higher viral load (lower ct-value) compared to matched immunocompetent controls (*p* = 0.08). The two kidney transplant recipients had the highest viral load (ct-value 21) in the whole cohort.

### 3.3. Longitudinal SARS-CoV-2 IgG Antibodies

At inclusion, four of six (67%) immunocompromised and seven of twelve (58%) immunocompetent patients had detectable levels of SARS-CoV-2 anti-N IgG antibodies, and two (33%) immunocompromised and seven (58%) immunocompetent patients had detectable anti-S IgG antibodies (Table 1). By two and six weeks, both anti-S IgG and anti-N IgG were present in all immunocompromised patients (Appendix A). At six months, anti-S IgG were detected in all patients whereas anti-N IgG, not boosted by the vaccine, was detected in only one (17%) immunocompromised patient in comparison with nine (75%) immunocompetent patients (Appendix A). The levels of anti-N IgG had decreased in the majority of samples. At six months, anti-S IgG ranged from 49 to 11,000 BAU/mL across the cohort. The highest levels of anti-S IgG were found in patients that had received at least one dose of vaccine at least four weeks before sampling (Appendix A).

### 3.4. Immunological Phenotype Six Months Post Infection

The group of immunocompromised patients was heterogenous regarding immunological dysfunction, which is reflected in their immunological profiles at six months post infection. Compared to immunocompetent patients, the lowest numbers of all lymphocyte subsets (B, CD4, CD8 and NK cells) were found among the immunocompromised (Figure 1a). However, on a group level there were no significant differences compared to the immunocompetent patients (Appendix A). At follow-up, vaccination against COVID-19 had been initiated; one immunocompromised and two immunocompetent patients had received one dose and two immunocompromised and five immunocompetent patients had received two doses of vaccine (Appendix A).

### 3.5. SARS-CoV-2 Specific Memory T Cells Six Months Post Infection

SARS-CoV-2-specific T cells were detected by a flow cytometric assay for specific T cell-mediated recall immune response by lymphoblast formation in activated whole blood [22]. Representative plots of the assay are shown in Appendix A. CD4 S-specific T cells were detected in all immunocompetent patients and in four of six immunocompromised patients. Spike peptide specific CD4 T cells were not detected in the two kidney transplant recipients despite recent vaccination against COVID-19 (Figure 1b). CD8 S-specific T cells were detected in one immunocompromised patient (anti-TNF therapy) and in two immunocompetent patients (data not shown). CD4 T cells specific for nucleocapsid peptide were detected in three immunocompromised and in nine immunocompetent patients (Figure 1b). CD8 N-specific T cells were detected in two immunocompromised and in one immunocompetent patient (results not shown). Overall, long lived N-specific T cell responses were detected in immunocompromised patients with normal or moderately decreased T cells.

### 3.6. Correlation of SARS-CoV-2 Specific T Cells with NK Cells

Nucleocapsid peptide specific CD4 and CD8 T cells detectable at six months post infection reflect long lived natural immunity to COVID-19. In the entire study cohort, including both the immunocompromised and immunocompetent patients, proliferating N-specific CD4 T cells correlated not only with the total number of CD4 T cells in whole blood, but also with the total number of NK cells. Neither the number of B cells nor CD8 T cells correlated with proliferating N-specific CD4 T cells (Figure 2). The number of S-specific CD4 T cells did not differ between vaccinated and unvaccinated patients (Figure 1b).

## 4. Discussion

During the first year of the pandemic, immunocompromised patients accounted for a relatively small proportion of hospitalized patients with COVID-19, as also shown in this study. The controls were immunocompetent, but instead had other well-known COVID-19 risk factors, such as ischemic heart disease, diabetes, and obesity. The median symptom duration and length of hospital stay did not differ between the groups. Interestingly, the highest viral load was detected in the two kidney transplant recipients who were on T cell inhibitory therapy. In fact, early activation of functional SARS-CoV-2 specific T cells are likely to be important for viral clearance, and consequently have a prognostic value in COVID-19 [23]. The biochemical variables and degree of inflammation did not differ, and the outcome of COVID-19 was the same in both groups, although one immunocompromised patient received intensive care.

All immunocompromised patients, of which none had received B cell depleting therapy, developed specific anti-N IgG and anti-S IgG antibodies during the first six weeks post infection. However, the anti-N antibody levels decreased significantly during the six- month follow-up period, and to a greater extent in the immunocompromised patients. This is in line with previously reported antibody kinetics for immunocompetent individuals [24,25]. Knowledge of antibody responses to SARS-CoV-2 in secondary immunodeficiency after recovery of COVID-19 is scant. A rapid decline of SARS-CoV-2 nucleocapsid antibodies has been reported in kidney transplant recipients, with more than 60% having negative or equivocal levels after six months [26]. In our study, anti-S IgG responses were found six months post infection in all patients vaccinated against COVID-19. IgG antibodies specific for the RBD of the spike protein have been shown to correlate with neutralization activity [27]. The level of neutralizing antibodies is predictive of protection against reinfection with SARS-CoV-2 [28]. However, the protective threshold of anti-S IgG has not been established. In this study, high levels of anti-S IgG (>300 BAU/mL) were detected in all vaccinated study participants six months post COVID-19, indicating protective immunity [21].

The presence of specific memory CD4 and CD8 T cells with proliferation capacity up to ten months after recovery from COVID-19 have been reported [29,30]. Specific memory T cell responses are usually studied in cultures of peripheral blood mononuclear cells isolated from whole blood. Instead, we used a flow cytometric assay for specific T cell recall responses in antigen stimulated whole blood [31,32]. This method has the advantage of computing both the number and function of cells, thus providing a measure of the total functional capacity, which is of clinical relevance. We found a correlation between total numbers of CD4 T cells and the number of nucleocapsid specific CD4 T cells, thus increased responsiveness may to a large extent be explained by adequate CD4 T cell numbers. In addition, the lack of difference in T cells specific for spike peptides between vaccinated and unvaccinated patients may reflect limitations of the vaccines to induce T cell responses. The detection of SARS-CoV-2-specific CD8 T cells was restricted to two immunocompromised patients with high CD8 T cell counts and two immunocompetent patients that had been vaccinated, which is in line with previous findings illustrating that COVID-19 vaccination elicits CD4 T cell dominant immune responses [33]. Memory T cells rapidly express effector activity after antigen recognition presented by non-professional antigen presenting cells. However, poor proliferation of CD8 T cells in vivo may also be the result of T cell exhaustion [34]. PD-1, which is known to mediate the functional impairment of memory CD8 T cell responses [35], is an immune checkpoint molecule, which has been shown to be upregulated by T cells in COVID-19 [36]. The strong CD4 and CD8 T cell responses seen in the two patients on a TNF-blocking regimen may be associated with the reduced TNF-receptor mediated apoptosis of T cells [37]. The use of TNF-inhibitors has been associated with a reduced risk of severe COVID-19 and augmented cellular vaccine responses [38,39].

NK cells contribute to the early anti-viral immune response against SARS-CoV-2 [40]. Peripheral blood NK cells are redirected from the circulation to the lungs in severe COVID-19 and display an activated cycling phenotype [41,42]. In addition, NK cell senescence has been suggested as an important factor in severe and fatal COVID-19 [43]. Thus, NK cell fitness may contribute to a mild disease course. In our small cohort, the number of N-specific CD4 T cells showed a positive correlation with whole blood NK cell numbers. NK cells influence antiviral T cell responses through a variety of mechanisms, e.g., by the production of interferon-γ that enhances T cell memory [44]. Moreover, NK cells can augment T cell recall responses by interaction with antigen presenting cells [45]. Recent studies show that NK cells can acquire immunological memory in a manner similar to B and T cells [46], and NK cell-mediated memory responses have been described in human cytomegalovirus infection [47]. It can be hypothesized that NK cells ensure a microenvironment in favor of robust recall anti-SARS-CoV-2 T cell responses, and that SARS-CoV-2 experienced NK cells are better regulators of anti-viral recall responses than SARS-CoV-2 naïve NK cells. Presently, there are ongoing clinical trials evaluating NK cell allogenic transfer as adjunctive treatment for COVID-19 in immunocompromised populations [48].

The main strengths of the study were the prospective, longitudinal design and the well-characterized patients. However, the small patient population and the heterogenous causes of immunosuppression were both obvious limitations of the study, as was the lack of lymphocyte profiles in the patients at all follow-up occasions.

## 5. Conclusions

The results of this pilot study provide some additional knowledge about the development of cellular immunity to SARS-CoV-2 in patients with diseases and therapies affecting the immune system. In certain immunosuppressive conditions and as a complement to the assessment of SARS-CoV-2 humoral immunity, lymphocyte phenotyping may convey important information by predicting the presence of long-lived SARS-CoV-2 specific T cell responses. Determining the durability of the humoral and cellular immune response against SARS-CoV-2 in immunocompromised individuals can provide important insights into the risk of re-infection, the need for vaccine boosters, and the need for pre- or post-exposure prophylaxis with SARS-CoV-2 neutralizing mAbs. Accordingly, the findings in this study need to be confirmed by studies with larger patient cohorts.

## Figures and Tables

**Figure 1 jcm-11-03535-f001:**
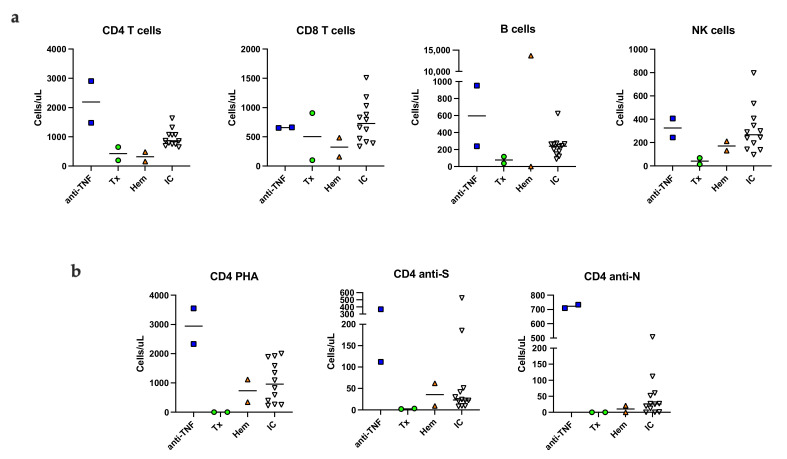
SARS-CoV-2 specific T cell responses in activated whole blood. Absolute number (cells/µL) of lymphocyte subsets in unstimulated peripheral blood (**a**). The number of PHA, S- and N-specific stimulated, blast transformed CD4 T cells (**b**). Anti-TNF; patients on TNF inhibitor, Tx; kidney transplant recipient, Hem; patients with myeloma or chronic lymphocytic leukemia, IC; immunocompetent patients, NK; natural killer cell, S; spike peptide, N; nucleocapsid peptide.

**Figure 2 jcm-11-03535-f002:**
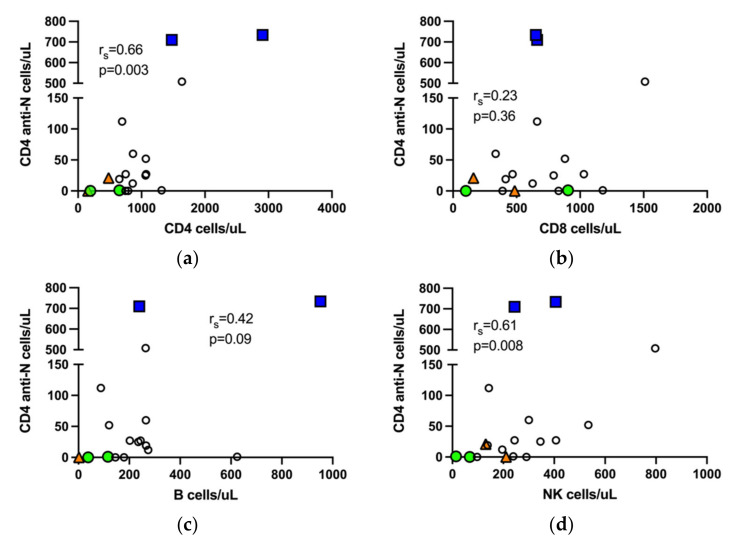
Nucleocapsid specific T cells in the entire study cohort correlate with CD4 T cells and NK cells. Correlation of N-specific CD4 T cells with total number of CD4 T cells (**a**), CD8 T cells (**b**), B cells, sample from a patient with chronic lymphocytic leukemia and B cell count >10,000 is not shown, (**c**) and NK cells (**d**), respectively, in peripheral blood. r_s_, Spearman correlation.

**Table 1 jcm-11-03535-t001:** Clinical characteristics of the study cohort, biochemical variables, and SARS-CoV-2 serology at inclusion.

	Immunocompromised *n* = 6	Controls *n* = 12
Sex, male, *n* (%)	3 (50)	6 (50)
Age, years, median (range)	62 (44–73)	64 (46–75)
Ever smoker, *n* (%)	3 (60)	7 (58)
Ischemic heart disease, *n* (%)	2 (33)	6 (50)
Lung disease, *n* (%)	2 (33)	3 (25)
Diabetes, *n* (%)	0	3 (25)
BMI, median (range)	26 (22–37)	30 (25–43)
Symptom duration at inclusion, days median (range)	12 (5–16)	10 (6–30)
Hospitalization, days median (range)	7 (2–54)	7 (3–15)
Intensive care ^#^, *n* (%)	1 (17)	0
Oxygen therapy, *n* (%)	3 (50)	11 (92)
HFNOT therapy, *n* (%)	2 (33)	5 (42)
Mechanical ventilation, *n* (%)	1 (17)	0
Remdesivir, *n* (%)	3 (50)	5 (42)
Dexamethasone, *n* (%)	4 (67)	8 (67)
**Biochemical variables at inclusion**		
Hemoglobin (g/L), median (range)	115 (87–126)	133 (98–154)
WBC (×10^9^/L), median (range)	6.1 (0.4–49)	6.9 (3.5–12)
Platelets (×10^9^/L), median (range)	154 (20–668)	282 (121–543)
Lymphocytes (×10^9^/L), median (range)	1.1 (0.3–33)	1.1 (0.3–2.8)
Neutrophils:lymphocytes, median (range)	2.1 (0.2–14.7)	4.0 (1.8–20)
C-reactive protein, (mg/L)	34 (11–376)	43 (7–258)
Plasma creatinine (umol/L,) median (range)	110 (54–341)	65 (51–188)
LDH (ukat/L), median (range)	6.3 (3.7–9.4)	6.8 (4.1–7.7)
**SARS-CoV-2 serology at inclusion**		
Anti-SARS-CoV-2 nucleocapsid IgG, *n* (%)	4 (67)	7 (58)
Anti-SARS-CoV-2 spike IgG, *n* (%)	2 (33)	7 (58)

^#^ The patient spent 24 days in the intensive care unit. BMI; body mass index, HFNOT; high-flow nasal oxygen therapy, WBC; white blood cell count; LDH; lactate dehydrogenase.

## Data Availability

The raw data supporting the conclusions of this article will be made available by the authors, without undue reservation.

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
