# Peer review of "SARS-CoV-2 Specific Antibody Response and T Cell-Immunity in Immunocompromised Patients up to Six Months Post COVID: A Pilot Study"

_jcm, 2022, doi:10.3390/jcm11123535_

Round 1
Reviewer 1 Report
The authors describe the development of both specific humoral and cellular responses to SARS-CoV-2 in a group of immunocompromised individuals, compared to a group of immunocompetent individuals. Being a prospective study, they followed the patients during the first 6 months with 2 and 6 weeks as intermediate time points. All patients developed IgG against N and S antigens but only S-IgG were maintained after 6 months (likely because of vaccination effect). Regarding T cell response, immunocompromised patients (4/6) were able to develop SARS-CoV-2-specific CD4+ immunity which correlated with the total number of CD4 T and NK cells. The relevance of this manuscript laid on the necessity of new data about SARS-CoV-2 specific immune responses in this kind of population, underrepresented in the current literature. However, as they mentioned in the manuscript, the low number of patients is a clear limitation along with the huge heterogeneity in immunosuppression that will clearly affect the results. Higher number of patients will undoubtedly improve the quality of the manuscript. However, vaccination and the current pandemic situation will make difficult to increase this number. I have some questions and suggestions for the authors.
- The study involves hospitalized patients with COVID-19. At inclusion, they performed IgG measurement. Why authors did not measure IgM? Are immunocompromised patients’ IgG levels comparable with those of immunocompetent individuals? Why this is not shown in Figure S2?
- The relevance of neutralizing antibodies have widely been shown. It would be interesting the evaluation of these antibodies and its durability in both groups.
- Only 5/6 patients had positive viral load at the inclusion moment. Was the negative patient confirmed by qPCR prior inclusion? Maybe authors might clarify that qPCR was performed at all patients at the moment of hospital admission.
- Might authors make reference to the cutoff value of the qPCR? What dotted line in Fig S2a is indicating?
- In line 168, they might remove the hyphen of CD4- and CD8- since this can be confused with negative cells.
- Might author specify whether the representative plots of Figure S3 come from an immunocompromised or immunocompetent individuals?
- Might author express their opinion about why anti-TNFa patients have such a different profile compared with the rest of patients?
Author Response
Response to Reviewer #1
The authors describe the development of both specific humoral and cellular responses to SARS-CoV-2 in a group of immunocompromised individuals, compared to a group of immunocompetent individuals. Being a prospective study, they followed the patients during the first 6 months with 2 and 6 weeks as intermediate time points. All patients developed IgG against N and S antigens but only S-IgG were maintained after 6 months (likely because of vaccination effect). Regarding T cell response, immunocompromised patients (4/6) were able to develop SARS-CoV-2-specific CD4+ immunity which correlated with the total number of CD4 T and NK cells. The relevance of this manuscript laid on the necessity of new data about SARS-CoV-2 specific immune responses in this kind of population, underrepresented in the current literature. However, as they mentioned in the manuscript, the low number of patients is a clear limitation along with the huge heterogeneity in immunosuppression that will clearly affect the results. Higher number of patients will undoubtedly improve the quality of the manuscript. However, vaccination and the current pandemic situation will make difficult to increase this number. I have some questions and suggestions for the authors.
Response: We are grateful for the feedback and agree with the comments regarding the limited study population in this brief report and therefore the results should be interpreted with caution. However, we believe that the study contributes to new knowledge and understanding of immunity assessment in immunocompromised patients with COVID-19. We address the views point-by-point below.
Question 1: The study involves hospitalized patients with COVID-19. At inclusion, they performed IgG measurement. Why authors did not measure IgM? Are immunocompromised patients’ IgG levels comparable with those of immunocompetent individuals? Why this is not shown in Figure S2?
Response to Q1: This is a relevant point. We assume that the reviewer refers to SARS-CoV-2 anti-spike IgM or anti-nucleocapsid IgM at study inclusion. Already early in the pandemic, the accredited microbiological laboratory at the University Hospital in Linköping (where the SARS-CoV-2 serology in the current study was performed) chose not to report IgM due to its low sensitivity, cross-reactive characteristics, and low specificity. In addition, IgM antibodies are not detected substantially earlier than IgG antibodies in COVID-19 patients. In fact, anti-spike IgG is often detected before a short-term IgM response in COVID-19 patients. This observation has been made by several research groups (a few examples are listed below):
- Rikhtegaran et al. PLoS One. 2020 Nov 2;15(11):e0237828;
- Long QX et al. Nat Med. 2020; 26(6):845–8).
- Latiano et al. Int J Infect Dis . 2021 Mar;104:159-163
- Elslande et al. Clin. Microbiol. Infect. 2020;26:1082–1087.
- Bouassa et al. J. Med. Virol. 2021;93:2196–2203.
- Vanroye et al. Diagnostics (Basel). 2021 Jul; 11(7): 1163.
We understand the reviewer’s concern that information of anti-SARS IgG of immunocompetent individuals is missing. We have added information by replacing Figure S2b. We show the dynamics of the IgG response in the whole study population. Anti-S IgG levels were evaluated with an assay that detects IgG against the receptor binding domain (RBD) of the spike protein and results are presented as binding arbitrary units (BAU)/mL.
In results line 170, the following text was added: in the majority of samples. At six months, anti-S IgG ranged from 49 to 11,000 BAU/mL across the cohort. The highest levels of anti-S IgG were found in patients that had received at least one dose of vaccine at least four weeks before sample collection (Fig S2b).
Question 2: The relevance of neutralizing antibodies have widely been shown. It would be interesting the evaluation of these antibodies and its durability in both groups.
Response to Q2: The reviewer has a good point in that we should better explain our interpretation of the presence of neutralizing anti-SARS-CoV-2 IgG.
Neutralizing antibodies to SARS-CoV-2 are directed against the receptor binding domain (RBD) of the spike protein, important for cellular entry by interaction with the ACE2 receptor. In this study we used a semiquantitative commercial chemiluminescent microparticle-based immune assay to detect anti-SARS-CoV-2 spike IgG, directed against the RBD (Abbott SARS-CoV-2 IgG II Quant/6S60 ARCHITECT SARS-CoV-2 IgG kit). The assay is calibrated to the WHO International Standard for anti-SARS-CoV-2 antibody (NIBSC Code 20-136). The anti-SARS-CoV-2 spike IgG are expressed as standardized binding antibody units (BAU)/mL (a conversion of AU using the WHO Standard). The assay has demonstrated linearly correlated results with plaque reduction neutralization test and predicts neutralization activity at titers above 17.0 BAU/mL. The cut-off for positive anti-RBD IgG used by our analyzing laboratory was 14.2 BAU/mL, thus anti-S levels correlate with neutralization activity in positive samples. Standard virus neutralization tests are laborious and take several days to perform. We think that the anti-RBD IgG assay adequately reflects the neutralization activity of the tested samples.
References:
Shang J, Wan Y, Luo C, Ye G, Geng Q, Auerbach A, Li F. Cell entry mechanisms of SARS-CoV-2. Proc Natl Acad Sci U S A. 2020 May 26;117(21):11727-11734. doi: 10.1073/pnas.2003138117. Epub 2020 May 6. PMID: 32376634; PMCID: PMC7260975.
Lee B, Ko JH, Park J, Moon HW, Baek JY, Jung S, Lim HY, Kim KC, Huh K, Cho SY, Kang CI, Chung DR, Huh HJ, Chung CR, Kim YJ, Joo EJ, Kang ES, Peck KR. Estimating the Neutralizing Effect and Titer Correlation of Semi-Quantitative Anti-SARS-CoV-2 Antibody Immunoassays. Front Cell Infect Microbiol. 2022 Apr 14;12:822599. doi: 10.3389/fcimb.2022.822599.
We have added the following information in methods:
Line 102:
Chemiluminescent microparticle-based immune assays were used for quantification of IgG directed against the receptor binding domain (RBD) of the spike protein (Architecht SARS-CoV-2 II Quant, 6S60)
Line 106-108:
The anti-S IgG RBD assay was calibrated to the WHO International reference anti-SARS IgG antibody NIBSC 20-136. Linear correlation with neutralization activity has been demonstrated for the anti-S IgG RBD assay at levels above 17 binding arbitrary units (BAU)/mL (Lee et al., Front Cell Infect Microbiol. 2022). The anti-SARS-CoV-2 S IgG positivity cut off was 14.2 BAU/mL. The anti-SARS-CoV-2 N IgG positivity cut off was 1.4 AU/mL.
The following changes have been made in results line 170:
The levels of anti-N IgG had decreased in the majority of samples. At six months, anti-S IgG ranged from 49 to 11,000 BAU/mL across the cohort. The highest levels of anti-S IgG were found in patients that had received at least one dose of vaccine at least four weeks before sample collection (Fig S2b). whereas spike specific IgG levels remained stable.
We have made the following changes in discussion line 238:
However, the anti-N antibody levels decreased significantly during the 6-month follow-up period, and to a greater extent in the immunocompromised patients.
Line 244
At follow-up, anti-S IgG responses were found in all patients vaccinated against COVID-19.
Anti-S specific IgG responses detectable for more than nine months were seen in most organ transplant recipients after recovery of COVID-19, suggesting a robust humoral response despite immune suppression [26]
IgG antibodies specific for the RBD of the spike protein have been shown to correlate with neutralization activity (Seow et al., Nat Microbiol. 2020). The level of neutralizing antibodies is predictive of protection of reinfection with SARS-CoV-2 (Khoury et al., Nat Med. 2021). However, the protective threshold of anti S IgG has not been established. In this study, high levels of anti-S IgG (> 300 BAU/mL) were detected in all vaccinated study participants at six months follow up, indicating protective immunity (Lee et al., Front Cell Infect Microbiol. 2022).
Question 3: Only 5/6 patients had positive viral load at the inclusion moment. Was the negative patient confirmed by qPCR prior inclusion? Maybe authors might clarify that qPCR was performed at all patients at the moment of hospital admission.
Response to Q3: Very relevant comment, thank you! As explained in the Materials and Methods section, at admittance to hospital, and as part of routine diagnostics, all patients and controls had COVID-19 verified by PCR using nasopharyngeal or throat swab specimens (inclusion criterium). Nasopharyngeal samples were resampled from patients and controls at study inclusion (at a later moment when admitted to the Department of Infectious Diseases, Vrinnevi Hospital, Norrköping), which explains why some of the study participants were PCR-negative. This has now been clarified in the Materials & Methods, section Real-time PCR for quantification of SARS-CoV-2 viral load, lines 80-86 and in the Results, section SARS-CoV-2 viral load, lines 149-151.
Question 4: Might authors make reference to the cutoff value of the qPCR? What dotted line in Fig S2a is indicating?
Response to Q4: In our previous in-house evaluations of the qPCR assay used, using SARS-CoV-2 positive samples as well as dilution series, we have observed the highest ct value for SARS-CoV-2 positivity be 38. Thus, the cut-off was set at 39, and all negative samples were given a value of 39 for statistical purposes (e.g. calculating median; non-parametric comparisons between groups). To note, the highest ct-value for samples in this study was 36.1 We have now modified the figure legend for clarity as well as added information in materials and methods section under “Real-time PCR for quantification of SARS-CoV-2 viral load”, last line. The dotted line in Fig S2a was indicating the start of qPCR cycling, i.e the minimum Ct value (0). It has now been removed for clarity.
Question 5: In line 168, they might remove the hyphen of CD4- and CD8- since this can be confused with negative cells.
Response to Q5: Corrected as suggested.
Question 6: Might author specify whether the representative plots of Figure S3 come from an immunocompromised or immunocompetent individuals?
Response to Q6: We apologize for missing to include this piece of information, and we have made an addition to the legend of figure S3. The data of the representative plot came from an unvaccinated immunocompetent individual with detectable anti-spike and anti-nucleocapsid IgG.
Question 7: Might author express their opinion about why anti-TNFa patients have such a different profile compared with the rest of patients?
Response to Q7: The distinct immunological profiles of the two patients on anti-TNF therapy is probably a result of T cell lymphocytosis, a phenomenon that has been described in rheumatoid arthritis and psoriatic arthritis patients on anti-TNF-blocking regimen (Picchianti Diamanti A, et al. J Transl Med. 2017). In addition, augmented T cell vaccine responses have been reported in IBD patients on anti-TNF therapy (Li D, et al., Infl Bowel Dis. 2022). It has been hypothesized that anti-TNF increases T cell expansion by blocking apoptotic signaling of the TNF-receptor expressed by activated T cells. Anti-TNF therapy is also associated with a reduced risk of severe COVID-19 infection (Izadi Z, et al., JAMA Netw Open 2021).
We have added the following text and references in Discussion line 271:
The strong CD4 and CD8 T cell responses seen in the two patients on TNF-blocking regimen may be associated with reduced TNF-receptor mediated apoptosis of T cells (Mehta A, et al., Cytokine 2018). Anti-TNF therapy has been associated with reduced risk of severe COVID-19 and augmented cellular vaccine responses (Izadi Z, et al., JAMA Netw Open 2021).
Reviewer 2 Report
In this manuscript, Sjöwall et al characterized and compared the immune response against SARS-CoV-2 infection between immunocompetent and immunocompromised individuals. By obtaining nasopharyngeal swabs and sampling blood over several time points, the authors report differences in viral loads, antibody titers against spike and nucleocapsid proteins and phenotypic components of the responding immune cells. The authors aim to address the durability and the quality of immune response by follow up the blood collection at 6 months post symptom onset. However, the observations of this study is severely limited by the total number of immunocompromised patients. Immune response in natural infection is humans is usually diverse as reported for several clinical infections. As the authors indicated, differences in the specific conditions behind immunocompromised may impact the onset and evolution of the immune response to the viral infection. Given the small number of patients in total and per condition, and the variation of immune response within a given cohort, the interpretation of the data presented in the manuscript remains to be validated and supported by performing similar analyses in more individuals.
Author Response
Response to Reviewer #2
In this manuscript, Sjöwall et al characterized and compared the immune response against SARS-CoV-2 infection between immunocompetent and immunocompromised individuals. By obtaining nasopharyngeal swabs and sampling blood over several time points, the authors report differences in viral loads, antibody titers against spike and nucleocapsid proteins and phenotypic components of the responding immune cells. The authors aim to address the durability and the quality of immune response by follow up the blood collection at 6 months post symptom onset. However, the observations of this study is severely limited by the total number of immunocompromised patients. Immune response in natural infection is humans is usually diverse as reported for several clinical infections. As the authors indicated, differences in the specific conditions behind immunocompromised may impact the onset and evolution of the immune response to the viral infection. Given the small number of patients in total and per condition, and the variation of immune response within a given cohort, the interpretation of the data presented in the manuscript remains to be validated and supported by performing similar analyses in more individuals.
Response: We are grateful for the feedback and agree with the comments regarding the limited study population in this brief report and therefore the results should be interpreted with caution and evaluated in studies with larger and homogeneous populations of immunosuppressed patients. Nevertheless, we believe that the study contributes new knowledge and understanding of immunity assessment in immunocompromised patients with COVID-19.
Reviewer 3 Report
This is a brief report article exploring both spike and nucleocapsid antibody response in hospitalized patients with COIVD-19, comparing immunocompetent and immunocompromised patients. This is a pilot study, and the sample size is small, but the authors acknowledge the limitation and submitted as a brief report format. I have some comments below:
1. The patient demographics between the two groups are important, and I recommend making Table S1 to Table 1. The demographics should be in the main text.
2. The authors should be clear about the definition of “immunocompetent,” Are they healthy before COVID-19 hospitalization? And also a clear definition of immunocompromised.
3. The information of couple of confoundings may affect the antibody response is lacking. For example, prior history of COVID-19 infection, time of admission (patients may have different virus variants), The vaccination type, time of vaccination, how many doses the patient received, the antibody measurement to the time of the last dose, days from symptoms onset to measurement, etc.
4. The first figure numbered as figure 2, but I cannot find figure 1 anywhere.
Author Response
Response to Reviewer #3
This is a brief report article exploring both spike and nucleocapsid antibody response in hospitalized patients with COIVD-19, comparing immunocompetent and immunocompromised patients. This is a pilot study, and the sample size is small, but the authors acknowledge the limitation and submitted as a brief report format.
Response: We appreciate the comments of the reviewer. We address the views point-by-point below.
Question 1: The patient demographics between the two groups are important, and I recommend making Table S1 to Table 1. The demographics should be in the main text.
Response to Q1: Corrected as suggested. Table S1 with patient demographics is now included in the main text as Table 1, and the current Table 1 is included in the Supplementary files as Table S1. Presenting anti-S as anti-RBD IgG in binding arbitrary units reduced the number of patients with positive serology at inclusion: Immunocompromised from 4 to 2, and immunocompetent from 9 to 7. These changes have been made to new Table 1. We have also made corrections accordingly in the body text, results line 164 and line 166. During re-evaluation of new Table 1, we noticed a couple of typos that also have been corrected.
Question 2: The authors should be clear about the definition of “immunocompetent,” Are they healthy before COVID-19 hospitalization? And also a clear definition of immunocompromised.
Response to Q2: This is of course a highly valid point and we have now clarified the definition of immunocompromised and immunocompetent patients in Materials & Methods, section Patients and blood sampling, lines 74-77. The immunocompetent patients presented with co-morbidities characteristic of hospitalized patients with COVID-19, such as cardiovascular disease, chronic pulmonary disease, diabetes, and obesity but without medical conditions or therapy directly affecting the immune system.
Question 3: The information of couple of confoundings may affect the antibody response is lacking. For example, prior history of COVID-19 infection, time of admission (patients may have different virus variants), The vaccination type, time of vaccination, how many doses the patient received, the antibody measurement to the time of the last dose, days from symptoms onset to measurement, etc.
Response to Q3: Thank you, relevant comment. The study was conducted during the second wave of the pandemic (August 2020 to May 2021 when the alpha and beta variants dominated), and none of the hospitalized patients had prior been tested positive for COVID-19. The antibody profiles of the inclusion samples indicated no previous exposure to COVID-19, since a secondary COVID-19 would have reactivated SARS-COV2 specific antibody memory responses with much higher antibody levels. Only two patients had anti-S IgG above 1000 BAU/mL at inclusion and one of these patients had a symptom duration of 23 days at the time of inclusion. The spike-antibody data in both immunocompromised and immunocompetent patients presented in new Figure S2b. In the 6 months follow-up samples, two immunocompromised patients and five immunocompetent patients were vaccinated (at least 1 dose more than 4 weeks before sample collection). Vaccination status is shown in the figure. Requested additional information, with individual patient data, is attached as a separate supplementary file.
Question 4: The first figure numbered as figure 2, but I cannot find figure 1 anywhere.
Response to Q4: We apologize for the confusion. Figure 2 has been rephrased to Figure 1, and Figure 3 accordingly to Figure 2.
Round 2
Reviewer 2 Report
I encourage the authors to word the claims more carefully given the limitation of sample numbers.
Author Response
Thank you for the comment! We agree that the results should be toned down, given the small patient cohort. In a revised version of the manuscript, we have accordingly made some changes, which are highlighted in yellow.

Reviewer 3 Report
The authors had addressed my comments
Author Response
Thank you. Much appreciated!